# Cardiovascular Inflammaging: Mechanisms and Translational Aspects

**DOI:** 10.3390/cells11061010

**Published:** 2022-03-16

**Authors:** Maria Luisa Barcena, Muhammad Aslam, Sofya Pozdniakova, Kristina Norman, Yury Ladilov

**Affiliations:** 1Department of Geriatrics and Medical Gerontology, Charité—Universitätsmedizin Berlin, Hindenburgdamm 30, 12203 Berlin, Germany; sofyapozdniakova@gmail.com (S.P.); kristina.norman@charite.de (K.N.); yury.ladilov@rub.de (Y.L.); 2DZHK (German Centre for Cardiovascular Research), Partner Site Berlin, 10785 Berlin, Germany; 3Experimental Cardiology, Department of Internal Medicine I, Justus Liebig University, Aulweg 129, 35392 Giessen, Germany; muhammad.aslam@physiologie.med.uni-giessen.de; 4Department of Cardiology, Kerckhoff Clinic GmbH, 61231 Bad Nauheim, Germany; 5DZHK (German Centre for Cardiovascular Research), Partner Site Rhein-Main, 61231 Bad Nauheim, Germany; 6Barcelona Biomedical Research Park (PRBB), Barcelona Institute for Global Health (ISGlobal), Doctor Aiguader, 88, 08003 Barcelona, Spain; 7Department of Nutrition and Gerontology, German Institute of Human Nutrition Potsdam-Rehbrücke, Arthur-Scheunert-Allee 114-116, 14558 Nuthetal, Germany; 8Department of Nutrition & Gerontology, Institute of Nutritional Science, University of Potsdam, Arthur-Scheunert-Allee 114-116, 14558 Nuthetal, Germany; 9Department of Cardiovascular Surgery, Heart Center Brandenburg, Brandenburg Medical School Theodor Fontane, University Hospital, Ladeburger Str. 17, 16321 Bernau, Germany

**Keywords:** cardiac inflammaging, vascular senescence, mitochondrial homeostasis, microbiome

## Abstract

Aging is one of the major non-reversible risk factors for several chronic diseases, including cancer, type 2 diabetes, dementia, and cardiovascular diseases (CVD), and it is a key cause of multimorbidity, disability, and frailty (decreased physical activity, fatigue, and weight loss). The underlying cellular mechanisms are complex and consist of multifactorial processes, such as telomere shortening, chronic low-grade inflammation, oxidative stress, mitochondrial dysfunction, accumulation of senescent cells, and reduced autophagy. In this review, we focused on the molecular mechanisms and translational aspects of cardiovascular aging-related inflammation, i.e., inflammaging.

## 1. Inflammaging

Chronic low-grade systemic inflammation is a well-established hallmark of aging [1,2]. It is characterized by high levels of circulating cytokines in the serum of older [3] but apparently healthy individuals [1,4,5], in the absence of general pathophysiological stress or acute infection. Chronic inflammation is involved in accelerated biological aging and age-related diseases, particularly cardiovascular diseases (CVD), type 2 diabetes, or cancer [6,7,8,9]. Accumulating data demonstrate an age-related increase in the levels of blood inflammatory markers such as tumor necrosis factor alpha (TNF-α), interleukin 1 beta (IL-1β), IL-6, and C-reactive protein (CRP). This condition is known as inflammaging [9]. Inflammaging is a sign and a cause of accelerated aging and a comprehensive marker of multimorbidity, disability, frailty, and premature death in old adults [9,10]. Moreover, it is associated with the failure of the immune system to clean pathogens and dysfunctional cells [6]. Inflammaging is driven by a variety of molecular age-related mechanisms, leading to cellular senescence [11], including impaired mitochondrial function, oxidative stress, DNA damage, inflammasome activation, and telomere shortening [11,12,13,14]. However, upregulation or activation of pro-inflammatory mediators plays a central role in inflammaging.

NF-κB is a central mediator of pro-inflammatory gene induction, and the upregulation of the NF-κB pathway has been documented in aging-related inflammatory disorders [15,16,17,18,19]. The well-established driver of the NF-κB-mediated inflammatory reaction is the formation of a Nod-like receptor pyrin domain containing 3 (NLRP3) inflammasome. NLRP3 is the most extensively studied inflammasome and is composed of NLRP3, apoptosis-associated speck-like protein containing a CARD (ASC), and pro-caspase 1. The NLRP3 inflammasome seems to play a pivotal role in several mechanisms related to inflammaging since inhibition of the NLRP3 inflammasome extends healthspan and diminishes age-dependent degenerative changes [20,21]. Increased NLRP3 inflammasome activity is associated with age-related pathological manifestations, including atherosclerosis, type 2 diabetes, and Alzheimer’s disease [20,22,23]. Within several downstream pro-inflammatory pathways activated by the NLRP3 inflammasome, IL-1β and IL-18 have a potential role for inflammaging [24]. Of note, IL-18 expression increases during the aging processes in humans [21,25,26]. In accordance with this, IL-18 is increased in aged healthy hearts in a sex-dependent manner [27].

Altogether, inflammaging is associated with several age-related diseases, including CVD, due to the aberrant release of several pro-inflammatory factors.

## 2. Cardiac Inflammaging

### 2.1. Low-Grade Systemic Inflammation in Aging Heart, Sex-Related Differences

Increased levels of circulating pro-inflammatory factors, e.g., CRP and IL-6, are closely associated with cardiovascular pathologies, including myocardial infarction or coronary heart disease [28,29,30,31].

Obesity, hypertension, diabetes, smoking, and atherosclerosis are risk factors for cardiac inflammaging. Recent research has also emphasized an essential role of biological sex in inflammaging-associated CVD. Loss of estrogen in older females promotes the activation of inflammatory pathways in cardiac aging [32] that are accompanied by a decline in antioxidative defense mechanisms and mitochondrial biogenesis and function [33,34]. Furthermore, non-diseased hearts from old women show reduced anti-inflammatory protection that is reflected by high levels of the pro-inflammatory cytokine IL-12 and low levels of the anti-inflammatory cytokine IL-10 (IL-12^high^/IL-10^low^) [27]. Interestingly, while cardiac inflammation becomes more prominent in aging female hearts, male hearts seem to be less prone to inflammation due to reduced activation of NF-κB in male hearts during aging [27]. In contrast to non-diseased hearts, NF-κB seems to be directly related to inflammatory processes in male myocardial tissue in age-related diseases such as dilated cardiomyopathy or end-stage myocarditis, since it is strongly upregulated in the diseased hearts solely in males [6,35,36,37].

In addition to the circulating pro-inflammatory cytokines contributing to cardiac aging, there is an increased amount of pro-inflammatory macrophages infiltrated into the myocardium of healthy older women [27], which are the most abundant resident immune cells in the heart [38]. We and others observed an age-related increase in the number of pro-inflammatory M1 macrophages that is accompanied by a decrease in the anti-inflammatory M2 macrophage phenotype in both healthy women and female mice [1,27]. In contrast to the healthy myocardium of old individuals, hearts from male aged patients with end-stage dilated cardiomyopathy and myocarditis show more CD68-positive macrophages ([35], unpublished data).

### 2.2. Role of AMPK and Sirtuins in Aging Heart

Cardiomyocytes are high energy-consuming cells, and a sufficient supply of ATP is essential for their contractile function. Involved in numerous signaling pathways controlling cellular metabolism, 5’ AMP-activated protein kinase (AMPK) is a key energy sensor and regulator of energy metabolism and mitochondrial homeostasis. ATP depletion activates AMPK, which leads to stimulation of catabolic processes to re-establish the energy balance [39]. In particular, AMPK switches off many energy-consuming processes, such as protein and lipid syntheses, and it activates energy-releasing processes, such as lipo- and glycolysis, autophagy, and fatty acid oxidation [40].

The mammalian heterotrimeric AMPK complex consists of a catalytic (alpha) subunit and regulatory (beta and gamma) subunits. The main, well-characterized mechanisms of AMPK activation are phosphorylation at Thr172 of the alpha subunit and allosteric modification by AMP and/or ADP binding to the gamma subunit. Under energy-depleted conditions, high levels of AMP and ADP bind to the AMPK gamma subunit, which prevents the phosphatases from accessing Thr172 of the AMPK alpha-subunit, thus increasing its phosphorylation. There are three upstream activating kinases that phosphorylate AMPK at Thr172: liver kinase B1 (LKB1), Ca^2+^/calmodulin-dependent protein kinase kinase 2 (CaMKK2), and TGF-β-activated kinase 1 (TAK1). The phosphorylated Thr172 can be dephosphorylated by the phosphatases protein phosphatase 1 (PP1), PP2A, and PP2C, resulting in AMPK inhibition. Moreover, there are several upstream Ser/Thr kinases, including PKA, AKT, and ERK1/2, that inactivate AMPK (for a review see [41,42]).

Increasing evidence suggests that the basal activity of cardiac AMPK, as well as its stress-related activation, decreases with age, although the total AMPK content remains mainly unchanged [27,42]. Although the precise underlying molecular mechanisms of age-related AMPK inactivation are not completely understood, the role of protein phosphatases, disturbance of the Ca^2+^ homeostasis, and activity of the inhibitory upstream kinases, e.g., PKA or ERK1/2, has been considered as a potential cause [42].

Another key cellular metabolic regulator is the deacetylase Sirt1, a member of the deacetylase family that utilizes NAD^+^ as a co-substrate to remove acetyl groups from lysine residues of a target protein. The protein deacetylation promotes enzyme-substrate binding and, therefore, enzyme activity. Notably, Sirt1 promotes AMPK activity via deacetylation and activation of AMPK’s upstream kinase LKB1 [43]. Several studies demonstrated that Sirt1 expression and activity declines with age in various tissues [44,45], which may also contribute to the impairment of AMPK signaling. Interestingly, we recently found that Sirt1 expression is significantly reduced in aged female but not aged male myocardium [27]. Accordingly, the acetylation of nuclear protein Ku70, a marker of the nuclear deacetylase activity, was significantly increased. Similar sex-dependent downregulation of another key cellular sirtuin, mitochondrial Sirt3, was also observed in this study. Sirt3 is a key mitochondrial deacetylase involved in regulating numerous mitochondrial enzymes, including superoxide dismutase 2 and isocitrate dehydrogenase 2 [46,47]. The net contribution of Sirt3 activity comprises the regulation of mitochondrial dynamics [48] and function [49]. The age-related downregulation of cellular deacetylase expression is further worsened by the impairment of sirtuin activity due to the reduced availability of NAD^+^ in aged cells. This phenomenon has been particularly attributed to the enhanced expression of the NAD^+^-consuming NADase CD38 in numerous tissues of old mice [50].

Age-related inactivation of key metabolic regulators such as AMPK and sirtuins has a significant negative impact on the cellular energy balance. In particular, mitochondrial function, biogenesis, and clearance control (mitophagy) are strongly dependent on the activity of AMPK, Sirt1, and Sirt3 [51,52,53]. Thus, the age-related decline in AMPK and sirtuin activity may lead to accumulation of dysfunctional and damaged mitochondria, which in turn leads to excessive ROS formation, the release of mitochondrial DNA (mtDNA) in the cytosol, and, as a result, inflammation [54,55] (see also below).

In addition, age-related downregulation of AMPK and sirtuin activity may promote cardiac inflammation in other ways. AMPK suppresses NF-κB activity via phosphorylation (at Ser177 and Ser181) and inactivation of IκB kinase (IKK), which attenuates activation of NF-κB signaling and the expression of pro-inflammatory genes [56]. Similarly, Sirt1 directly represses NF-κB gene expression by deacetylating RelA/p65 at lysine 310 [37]. Furthermore, AMPK suppresses NF-κB signaling indirectly via several downstream mediators, e.g., Sirt1 or PGC-1α, which can subsequently reduce the expression of pro-inflammatory factors [57]. Our recent report [27] demonstrated the association of age-related AMPK inactivation and cardiac inflammation in human heart. In agreement, activation of AMPK attenuates inflammation [58,59].

Similarly, downregulation of Sirt1 activity may promote cardiac inflammation. Indeed, Sirt1 promotes mitochondrial biogenesis and autophagy via deacetylation of PGC-1α [60], a key transcription factor regulating the expression of mitochondrial proteins, and Sirt1 increases AMPK activity via LKB1 deacetylation [43]. Thus, age-related Sirt1 downregulation may impair mitochondrial function, leading to mitochondrial ROS formation and mtDNA release.

Finally, downregulation of AMPK and Sirt1 leads to a metabolic shift that arises from the downregulation of mitochondrial oxidation of substrates and upregulation of glycolytic pathways. This shift to glycolysis may promote an inflammatory reaction [61,62] and, thus, may additionally contribute to disease development and age-related inflammation.

Taken together, the current knowledge argues for a significant role of the age-related downregulation of key metabolic regulators (AMPK, Sirt1, and Sirt3) in cardiac inflammation.

### 2.3. Mitochondria and Cardiac Inflammaging

Mitochondrial dysfunction is viewed as one of the major hallmarks of aging [6]. Cardiac aging is associated with the general decline in mitochondrial function and accumulation of dysfunctional mitochondria, mainly due to the dysregulation of quality control processes [63,64]. Mitochondria are a key source of the reactive oxygen species (ROS), which not only serve as signaling molecules but may also be destructive: accumulation of ROS promotes enhanced oxidative stress that gives rise to subsequent accumulation of damaged DNA, proteins, and lipids as well as mtDNA damage and release [65]. Although ROS are key signaling messengers required for proper cell functioning, when their exaggerated production level exceed the capacity of ROS scavenger systems that neutralize them, ROS become harmful to the cell. Indeed, we and others have clearly seen diminished antioxidant expression in aging individuals [27,66], thus explaining age-associated damage caused by ROS.

In old individuals, ROS can significantly increase the release of mtDNA from the mitochondrial matrix into the cytosol of a cell [67], which produces damage-associated molecular patterns (DAMPs) and is considered to be a driver of inflammatory responses [68]. Released mtDNA may stimulate many pattern recognition receptors, leading to activation of the cyclic GMP–AMP synthase-stimulator of interferon genes protein (cGAS-STING) pathway and hyperactivation of the innate immune response through Toll-like receptor (TLR) signaling; here, activation of the NLRP3 inflammasome play a central role (detailed review [69,70]).

Due to its proximity to a major source of ROS, mtDNA frequently harbors numerous mutations (for mtDNA mutation refer to a comprehensive review [71]). Thus, as each mitochondrion has several mtDNA copies, some of them may be mutated, while others are intact—the so-called heteroplasmy phenomenon that is typical for mitochondria [72]. The level of the heteroplasmy is increased during the aging processes, and mitochondria bearing mtDNA heteroplasmy is a hallmark of aging [73]. Moreover, aging-related mutations in mtDNA cause disruption of cellular homeostatic mechanisms and mitochondrial dysfunction via impaired oxidative phosphorylation [74].

Mitochondria are dynamic organelles whose function is intimately linked to their morphology; this is regulated by opposing fusion and fission, which are essential processes not only for the division but also for the preservation of functional mitochondrial integrity [75]. A disturbance of the balance between fusion and fission promotes the accumulation of damaged mitochondria and a fragmented mitochondrial network. Thus, cardiac aging is accompanied by disrupted mitochondrial structure and expanded mitochondrial size [76].

## 3. Vascular Inflammaging

CVD is the main cause of death worldwide [9], with increasing incidence in aged individuals. Enhanced vascular aging is associated with more severe atherosclerosis and microvascular dysfunction [77,78], and it is characterized by pathological vascular remodeling and vascular stiffness [79].

Inflammation is typical for aging hearts [27]. Within numerous factors promoting cardiac inflammation, circulating pro-inflammatory factors, which are highly elevated in old individuals play a substantial role [21].

Accumulating data demonstrate that vascular aging starts as early as childhood and is characterized by gradual changes in the vascular structure (e.g., luminal dilation and intimal and medial thickening) [80] and function (e.g., endothelial dysfunction), resulting in reduced vascular compliance and increased arterial stiffness [81]. The major hallmarks of vascular aging include impaired endothelium-dependent vasodilation and defective vessel repair capacity. Understanding the mechanisms mediating vascular aging may allow specific pathways to be targeted in order to delay the progression and adverse outcome of vascular aging.

The endothelium maintains normal vascular tone via releasing several vascular protective factors, such as endothelial-derived hyperpolarizing factor (EDHF) and nitric oxide (NO), under normal homeostatic conditions [82]. Endothelial dysfunction is the major hallmark of cardiovascular aging and is defined as the failure of endothelium to mediate an adequate vasodilatory response to hemodynamic stimuli such as shear stress [83,84]. Under these conditions, there is a reduction in the bioavailability of vasodilators (particularly NO) but an increase in endothelial-derived contracting factors [85,86]. Moreover, a dysfunctional endothelium is associated with a pro-inflammatory and pro-thrombotic state as well as with an increased risk of cardiovascular events [87,88,89].

Even in the absence of other cardiovascular risk factors, aging progressively causes reduced NO bioavailability [85,90] and endothelial dysfunction [91,92]. In endothelial cells, NO is produced via conversion of L-arginine to L-citrulline catalyzed by endothelial NO synthase (eNOS) in the presence of nicotinamide adenine dinucleotide phosphate (NADPH), tetrahydrobiopterin (BH4), and other cofactors [93,94]. In endothelial dysfunction, the reduced bioavailability of NO may be the result of reduced expression or activity of eNOS, reduced supply of eNOS substrate (L-arginine), increased endogenous eNOS inhibitors, or increased NO scavenging. Indeed, reduced eNOS activity accompanied by an impaired vasodilatory response to acetylcholine and bradykinin was observed in the vessels of aging rats [95,96]. Likewise, flow-mediated vasodilation was impaired in aged rats, which was ameliorated by hydralazine treatment [97].

Arginase, an important enzyme of the urea cycle, competes with eNOS for L-arginine as substrate and, thus, may limit the availability of L-arginine for NO production, particularly under conditions of increased arginase activity. An upregulation of arginase expression as well as activity that was accompanied by reduced vasodilation has been demonstrated in aged rat aortas; pharmacological inhibition of arginase improved the vasodilatory response [98]. Similarly, in rabbit cavernous, the carbachol-induced vasodilatory response was impaired in aged animals along with upregulation of arginase activity, and this impaired vasodilation was normalized by treatment with arginase inhibitors or excessive supply of L-arginine [99]. Indeed, arginase inhibition reverses endothelial cell aging phenotype in vitro [100]. Excessive production of ROS in the vascular wall may also reduce NO bioavailability by converting it to peroxynitrite [101,102,103]. This was verified by the observation that aging-induced reduction in NO bioavailability in rat vasculature was improved by administering the ROS scavenger TEMPOL [104]. Reduced vascular NO bioavailability may also result from a deficiency in BH4, an important cofactor for enzymatic NO production. BH4 depletion leads to eNOS uncoupling, resulting in increased ROS instead of NO production [105]. Indeed, reduced levels of BH4 accompanied by impaired endothelium-dependent vasodilation has been observed in arterioles of aged rats [106]. Supplementation of BH4 in these rats improved flow-mediated vasodilatation. Likewise, BH4 supplementation improved flow-dependent vasodilation in sedentary but not exercising older human adults [107].

The factors discussed above create a pro-inflammatory environment in the vessel wall and perivascular region that causes of infiltration of inflammatory cells mainly monocytes and macrophages [108]. For example, in aged Ldlr(−/−) mice on a high fat diet, the infiltration of monocytes and macrophages in aortic tissue was much higher compared to young littermates [109]. Likewise, number of perivascular infiltrating monocytes and macrophages in aging hypertensive rats was much higher compared young rats [110,111]. In addition to monocytes and macrophages, some recent reports indicate the involvement of other immune cells in eliciting aging-related tissue inflammation. For example, it has been reported that aging in mice is associated with increased accumulation of CD4^+^-T cells with dysfunctional mitochondria in mediastinal lymph nodes where they secrete massive amounts of IFN-γ [112,113], which plays crucial role in myocardial aging [114].

## 4. Inflammaging and the Microbiome

The microbiome contributes substantially to health, and as such, is also an important modifier of disease. Although definitions differ slightly [115], the microbiome is understood to be the community of all microorganisms such bacteria, viruses, fungi, and protozoa living in or on the human body and interacting with it. While most of the bacteria colonize the gastrointestinal tract, making the gut microbiome the most established and studied microbiota, bacteria also reside in the oral cavity, the skin, the vagina, and the urinary tract [116]. The effects of aging on the gut microbiota, as discussed below, have been well established, but the non-gut microbiome is also known to change with aging, which in turn has implications for organ health and may result in various diseases [117,118].

The microbiome is affected by various factors in higher age, including poly-medication and certain lifestyle factors, such as decreased physical activity and impaired dietary intake. Aging itself is known to impact many aspects of the gut and may also impair the functionality of the gastrointestinal tract. Disturbed motility, a decrease in digestive enzymes, and a prolongation of the colonic transit time, as well as increased intestinal permeability, lead to a functional decline of the aging gut [119].

Immune barriers have been reported to be compromised in higher age. For example, decreased production of antimicrobial mucin in the epithelial cells, which constitute a first important barrier, may lead to higher permeability and facilitate bacterial translocation [120]. The increased uptake of lipopolysaccharides into the bloodstream and further extra-intestinal sites may in turn trigger pro-inflammatory processes [121].

Concomitantly, intestinal microbiota can be drastically altered in advanced age, resulting in an aged-type microbiota [122] that exhibits a loss of biodiversity, an increase in opportunistic pathogens, and a reduction of health-associated species, such as short chain fatty acid (SCFA)-producing species [123]. In addition to their role in energy production, SCFA such as acetate, propionate, and butyrate have many health-promoting and protective properties, such as anti-inflammatory [124] and immunomodulatory properties [125,126]. Their role in the regulation of inflammation is well established [124]. SCFAs are produced by the intestinal tract during anaerobic fermentation of indigestible fibers and resistant starch. They, in particular butyrate, can trigger signaling cascades via activation of G-protein-coupled receptors (GPCRs) such as GPR41, GPR43, and GPR109A, and thus, exert an important immunomodulatory role, which has been documented in intestinal inflammation [127]. Due to these immunomodulatory functions and the impact of dysbiosis on low-grade systemic inflammation, recent evidence also points to a close relationship between the microbiome and inflammaging [128]. Animal studies have shown that, typically, age-related changes in the gut microbiome result in a pro-inflammatory status, with increased levels of IL-6, IL-10, TNF-α, and TGF-β as well as activation of NF-κB and mTOR and decreased levels of cyclin E and CDK2. In older humans, an increase in proteobacteria and a decrease in butyrate-producing species were associated with the increased levels of IL-6 and IL-8 [129]. A recent systematic review summarizing the evidence also revealed associations between certain bacteria (such as *Parabacteroides*, *Mucispirillum*, *Clostridium*, and *Sarcina*) and the pro-inflammatory cytokine MCP-1, whereas *Lactobacillus*, *Akkermansia*, *Oscillospira*, and *Blautia* are negatively associated with MCP-1 [130]. Moreover, transfer of aged type microbiota from old mice to germ-free mice resulted in increased systemic inflammation, as inflammation in the gut was promoted with subsequent transfer of lipopolysaccharides and increased T cell activation [131], which implies a causal role of the gut microbiome in inflammaging.

Age-related changes in the microbiome composition (age-related gut dysbiosis) have also been associated with the development of the frailty syndrome and anorexia of aging, which in turn may trigger further catabolic processes, although more studies are needed in this context. Gut microbiome dysbiosis has, moreover, been implicated in the pathophysiology of many diseases, and emerging evidence has also clearly established an association between the gut microbiome and the development of cardiovascular disease [132,133,134,135], especially via inflammatory pathways triggered by the interaction between host and microbiota [136], so that microbiome-based strategies for prevention of cardiovascular disease have been proposed [137]. However, causative mechanisms have not thus far been fully elucidated [137]. A role has been suggested for the gut microbiota-derived marker trimethylamine N-oxide (TMAO), a pro-atherogenic metabolite. It is an oxidation product of trimethylamine derived from bacterial metabolism of dietary L-carnitine, choline, and betaine [138]. High plasma concentrations of TMAO have previously been associated with a higher risk of cardiovascular disease [139] and cardiovascular mortality in peripheral artery disease [140].

While causality regarding the role of the microbiome in disease development is still a matter of debate, studies with fecal transplantation have shown that transplanted microbiota of adults with diseases such as cardiometabolic syndrome [141], hypertension [142], or obesity [143] have the potential to impact metabolism, inflammation, and body composition phenotypes in their host environments, which clearly implies a causal relationship between the microbiome and disease development. Similarly, in preclinical research, fecal microbiota transplantation decreased inflammation and, thus, alleviated myocardial injury in an experimental autoimmune myocarditis mouse model [144]. The use of nanomedicine in order to modify gut microbiota for the prevention of coronary artery disease is currently under investigation; it has been shown that nanoparticles can be used to transfer specific gut microbiota that are associated with a decrease in inflammation as well as an increase in SCFA and HDL [137].

Dietary quality, i.e., the quantity and quality of nutrients, also has a direct effect on the gastrointestinal microbiome, which in turn partly explains the beneficial effect of certain dietary patterns. As such, recent approaches in nutritional intervention have specifically targeted the aged-type microbiome. A one-year intervention with an individualized Mediterranean diet in older adults has been shown to successfully modify the microbiome and subsequently decrease inflammation and improve the aging phenotype frailty, clearly outlining the potential of dietary modification of aged-type microbiota [145].

## 5. Translational Aspects of Cardiovascular Inflammaging

Several large-scale studies, including those of the MARK-AGE group and NU-AGE consortium, the Leiden Longevity Study, and the “Berliner Alterstudie II”, have identified biomarkers of aging and inflammaging [31]. These studies revealed that CRP, TNF-α, and IL-6 are tentative candidate parameters of systemic inflammation related to aging [146,147,148,149].

Low-grade systemic inflammation is associated with mitochondrial dysfunction and biogenesis. Preservation of mitochondrial morphology, dynamics, and function may be a therapeutic approach to prevent cardiac aging. There are several potential drugs that may improve age-related mitochondrial dysfunction and, thus, attenuate cardiac inflammaging. One of them that has been tested in large clinical trials to combat age-related disorders and improve healthspan is metformin, which has a therapeutic effect by lowering oxidative stress via mitochondrial complex I inhibition, followed by the increase in cytoplasmatic AMP:ATP and ADP:ATP ratios, which in turns leads to activation of AMPK [150].

Similar to metformin, some nutraceuticals, i.e., agents derived from natural sources, possess a potential to interfere inflammaging via activation of cAMP-AMPK signaling [151]. Particularly, resveratrol, a stilbenoid produced by several plants, has been intensively investigated in numerous clinical trials related to various diseases [152]. A pioneer study by Park et al. [153] reported in skeletal muscle cells that resveratrol acts as a PDE inhibitor, enhancing cAMP levels and leading to activation of AMPK in an EPAC1-dependent manner. The authors showed that activation of EPAC1 increases intracellular Ca^2+^ levels and promotes CaMKK2 activity, which phosphorylates and activates AMPK. In line with this study, resveratrol attenuates endothelial inflammation through the activation of the cAMP-PKA-AMPK-SIRT1 signaling pathway [154].

Another potential approach to protect heart against inflammaging is attenuation of oxidative stress. Supplementation with coenzyme Q10, which naturally decreases with age [155], protects the heart from aging-related oxidative stress and improves mitochondrial function [156] by inhibition of mtDNA release and its accumulation in cytosol [157]. In a double-blind trial, long-term coenzyme Q10 treatment reduced major adverse cardiovascular events in heart failure patients [158]. There is an ongoing clinical trial in a large study population designed to confirm clinical benefits of long-term treatment with coenzyme Q10 in patients with cardiovascular disorders (NCT03133793). While these treatments are not mitochondria specific, there are some approaches targeting mitochondria that utilize triphenylphosphonium as a mitochondria-targeted vehicle to deliver antioxidants such as mitoquinone, SkQ1, or Mito-Tempo, to mitochondria [159]. The preclinical studies demonstrated promising beneficial results in patients with cardiac disorders, and there are ongoing clinical trials of these treatments (for a comprehensive review, see [159]).

In conclusion, cardiac inflammaging leads to several dysfunctional processes, including energy imbalance, mitochondrial dysfunction, changes in the microbiome, and vascular senescence. The prevention of cardiac aging and age-related CVD might be focus by the modulation of inflammaging and its related mitochondrial dysfunction (Figure 1).

## Figures and Tables

**Figure 1 cells-11-01010-f001:**
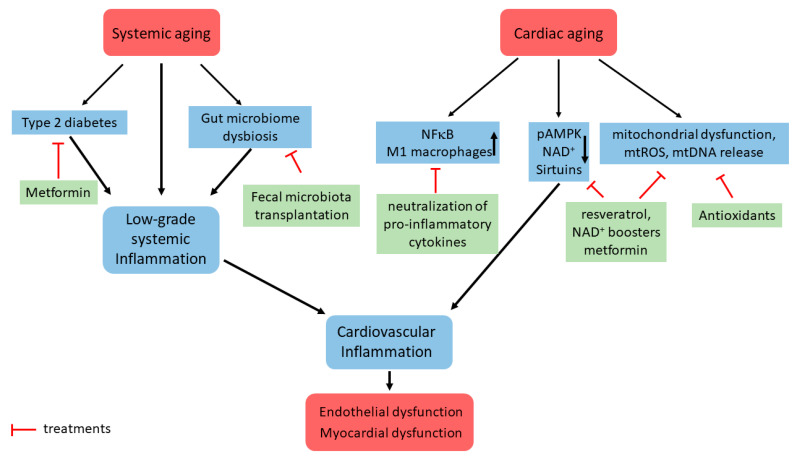
Schematic representation of some signaling pathways involved in cardiac inflammaging development and potential treatment strategies. Systemic aging leads to low-grade systemic inflammation, which among numerous other mechanisms includes type 2 diabetes and gut microbiome dysbiosis. Cardiac aging is linked with (i) increased expression of pro-inflammatory factors and immune cell infiltration, (ii) downregulation of main energy regulating mechanisms, such as pAMPK and sirtuin, and (iii) mitochondrial dysfunction. Together with the low-grade systemic inflammation, it leads to cardiac inflammation and endothelial/myocardial dysfunction. The potential treatment strategies are shown.

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
