# Peer review of "Cardiovascular Inflammaging: Mechanisms and Translational Aspects"

_cells, 2022, doi:10.3390/cells11061010_

Round 1

Reviewer 1 Report

Barcena et al. present a review dealing with the role of inflammation pathways in cardiovascular aging. The objective of this review is to gather many of the numerous findings published on the topic, going from mechanistic to clinical perspectives. The bibliography is well documented and the review represents an up-to-date account on the role inflammaging in heart and vessels.

Here are some specific comments :

  • In the abstract, the authors say : “In this review we focused on the molecular mechanisms and translational aspects of cardiovascular aging” (last sentence). However, aging is not really the topic of this review. Instead, it is mostly focus on inflammaging. The authors should change the last word of this sentence to “inflammaging” instead of “aging”.
  • The first paragraph called “1. inflammaging” should be more general and should deal with the role of inflammaging in the context of general aging. It should better introduce what is inflammaging. It should not be focus on the vessels, since there is a dedicated part called “3. Vascular inflammaging” in the review.
  • The authors could provide more detail information on the type of inflammatory cells that are present in the old vessels and old hearts (macrophages, lymphocytes…). Which cells release the cytokines ? What is their origin (infiltrating or resident ?)
  • The authors should consider rewriting the sentence in paragraph 1: “Within several downstream effects of NLRP3 inflammasome activation of pro-inflammatory IL-64 1β and IL-18 through cleavage by caspase-1 has a potential role for inflammaging (48)”

Author Response

In the abstract, the authors say: “In this review we focused on the molecular mechanisms and translational aspects of cardiovascular aging” (last sentence). However, aging is not really the topic of this review. Instead, it is mostly focus on inflammaging. The authors should change the last word of this sentence to “inflammaging” instead of “aging”.

  • We have revised that accordingly. We change into ...cardiovascular-related inflammation, i.e.- inflammaging (Line 28-29).

 The first paragraph called “1. inflammaging” should be more general and should deal with the role of inflammaging in the context of general aging. It should better introduce what is inflammaging. 

  • We have revised that accordingly. Moreover, it is associated with the failure of the immune system to clean pathogens and dysfunctional cells. Inflammaging is driven by a variety of molecular age-related mechanisms leading to cellular senescence.... (Line 42-45).

It should not be focus on the vessels, since there is a dedicated part called "3. Vascular inflammaging" in the review.

  • We have revised that accordingly. We excluded the vessels from the first section (1. Inflammaging).

The authors could provide more detail information on the type of inflammatory cells that are present in the old vessels and old hearts (macrophages, lymphocytes…). Which cells release the cytokines ? What is their origin (infiltrating or resident ?)

  • We have revised that accordingly. We changed into: In addition to the circulating pro-inflammatory cytokines contributing to cardiac aging, there is an increased amount of pro-inflammatory macrophages infiltrated into the myocardium of healthy older women, which are the most abundant resident immune cells in the resting heart. (Line 87-89).

The authors should consider rewriting the sentence in paragraph 1: “Within several downstream effects of NLRP3 inflammasome activation of pro-inflammatory IL-1β and IL-18 through cleavage by caspase-1 has a potential role for inflammaging (48)”

- We rewrite the sentence: Within several downstream pro-inflammatory pathways activated by the NLRP3 inflammasome, IL-1β and IL-18 have a potential role for inflammaging. (Line 60-61)

Reviewer 2 Report

Review article “Cardiovascular Inflammaging: Mechanisms and Translational 2 Aspects” (cells-1603588).

In this review authors comment on molecular mechanisms and translational aspects of cardiovascular aging. Manuscript is well written and easy to read. I would only suggest to check sentence lines 74/75, “There is a close association between increased levels of circulating pro-inflammatory factors, e.g., CRP and IL-6 (with?) myocardial infarction, or coronary heart disease”. Written as it is, it may be confusing to some readers.

Subject is scientifically relevant and interesting for audience working on cardiovascular and aging research, drawing attention to few important aspects of inflammaging, including not so often discussed microbiome.

In Figure 1, the potential treatment strategies includes agent resveratrol which is not, like others, mentioned in the main text. Therefore, I suggest to add at least short description/explanation of resveratrol in the mail text as well.

Author Response

In this review authors comment on molecular mechanisms and translational aspects of cardiovascular aging. Manuscript is well written and easy to read.

I would only suggest to check sentence lines 74/75, “There is a close association between increased levels of circulating pro-inflammatory factors, e.g., CRP and IL-6 (with?) myocardial infarction, or coronary heart disease”. Written as it is, it may be confusing to some readers.

  • We have revised that accordingly. We write the sentence: Increased levels of circulating pro-inflammatory factors, e.g., CRP and IL-6 are closely associated with cardiovascular pathologies including myocardial infarction, or coronary heart disease. (Line 70-71)

In Figure 1, the potential treatment strategies includes agent resveratrol which is not, like others, mentioned in the main text. Therefore, I suggest to add at least short description/explanation of resveratrol in the mail text as well.

- We have revised that accordingly. We included now resveratrol: Similarly to metformin, some nutraceuticals, i.e., agents derived from natural sources possess a potential to interfere inflammaging via activation of cAMP-AMPK signaling (151). Particularly, resveratrol, a stilbenoid produced by several plants, has been intensively investigated in numerous clinical trials related to various diseases (152). A pioneer study by Park et al (153) reported in skeletal muscle cells that resveratrol acts as a PDE inhibitor, enhancing cAMP levels and leading to activation of AMPK in an EPAC1-dependent manner. The authors showed that activation of EPAC1 increases intracellular Ca2+ levels and promotes CaMKK2 activity, which phosphorylates AMPK. In line with this study, resveratrol attenuates endothelial inflammation through the activation of the cAMP-PKA-AMPK-SIRT1 signaling pathway (154).